# Olfactory Ensheathing Cells Alleviate Facial Pain in Rats with Trigeminal Neuralgia by Inhibiting the Expression of P2X7 Receptor

**DOI:** 10.3390/brainsci12060706

**Published:** 2022-05-30

**Authors:** Jiafeng Lu, Baolin Yang, Jiayi Liao, Baokang Chen, Mingxin Lu, Wenjun Zhang, Jingnan Zeng, Hui Cheng, Zengxu Liu

**Affiliations:** 1Department of Anatomy, Basic Medical School, Nanchang University, Nanchang 330006, China; 401442720040@email.ncu.edu.cn (J.L.); yangbaolin@ncu.edu.cn (B.Y.); 406400210023@email.ncu.edu.cn (J.Z.); 406400210011@email.ncu.edu.cn (H.C.); 2The Stomatology Medical College of Nanchang University, Nanchang 330006, China; 6300317070@email.ncu.edu.cn; 3The First Medical College of Nanchang University, Nanchang 330006, China; chenbk366@163.com; 4The Queen Mary College of Nanchang University, Nanchang 330006, China; 6300517178@email.ncu.edu.cn; 5Second Affiliated Hospital of Nanchang University, Nanchang 330006, China; zwj880904@163.com

**Keywords:** cell transplantation, chronic constrictive injury, glial cell, immunofluorescence, infraorbital nerve, neuropathic pain, polymerase chain reaction, Sprague−Dawley, trigeminal ganglia, trigeminal nerve

## Abstract

Trigeminal neuralgia (TN) is a common facial neuropathic pain that is mainly characterized by spontaneous or induced needling or electric shock pain in the innervation area of the trigeminal nerve. It is also referred to as “the cancer that never dies”. The olfactory ensheathing cell (OEC) is a special glial cell in the nervous system that has a strong supportive function in nerve regeneration. Cell transplantation therapy is a useful treatment modality that we believe can be applied in TN management. In this study, OECs were transplanted into the ligation site of the infraorbital nerve of rats. We found that after the OEC transplantation, mechanical pain threshold in the face of the rats was significantly increased. Western blotting, immunofluorescence assay, and reverse transcription-quantitative polymerase chain reaction were performed on the trigeminal ganglia (TG) of model rats. The results revealed a decrease in the expression of P2X7 receptor (P2X7R) in the trigeminal ganglia. Our findings show that OEC transplantation has a good therapeutic effect on TN in rats, and that can reduce the expression of P2X7R in trigeminal ganglia. Therefore, we think that OEC transplantation may be a suitable treatment for TN.

## 1. Introduction

Trigeminal neuralgia (TN) is a common facial neuropathic pain that mainly manifests as spontaneous or induced needling or electric shock pain in the innervation area of the trigeminal nerve. The pathogenesis of TN is still unclear; however, it is proposed to involve demyelination and microvascular compression [1,2]. Moreover, the neural short circuit theory plays an important role in the various pathogenetic mechanisms proposed to be associated with TN. According to the theory, demyelination of the trigeminal nerve results in contact between adjacent exposed axons as well as nociceptive and non-nociceptive fibers, and transmission of spontaneous or heterotopic impulses through ephapses. These changes cause the nerve center to recognize non-pain signals as pain signals, which results in the sensation of pain [3,4,5]. Clinical treatment of TN mainly involves drug therapy and surgery. Currently, the drugs used for long-term therapy have a poor therapeutic effect. Additionally, the surgical treatment is associated with common sequelae such as facial numbness and paresthesia [6,7,8].

Olfactory ensheathing cells (OECs), the most myelinated cells, are the sheath blasts of the olfactory nerve that aid in nerve regeneration throughout life. The role of OECs in axon regeneration is unclear; however, OECs can secrete several neurotrophic factors including brain-derived nerve growth factor (BDNF) and glial cell line-derived nerve growth factor (GDNF) [9,10]. Cellular analgesia in animals is achieved through cell transplantation into the central nervous system and alleviation of sustained pain by secreting bioactive analgesics. This method has several application prospects in many cells with excellent properties. OECs have been used in the clinical and preclinical trials of spinal cord injury and have achieved good therapeutic outcomes [11,12,13].

Purinergic P2X1-P2X7 receptors (P2X1R-P2X7R) are adenosine triphosphate (ATP) channel regulatory proteins. P2X receptors are involved in the occurrence and development of pain [14]. Pain and other injurious stimuli induce cells to secrete excess ATP, which acts as an energy storage substance in the human body and a signaling factor for purinergic receptors. ATP and the purinergic receptors are mainly involved in the transmission of pain and nociceptive stimuli in primary sensory neurons [14,15,16,17]. The purinergic receptors P2X4R and P2X7R play an important role in this process. A large amount of evidence shows that P2X4R in microglia play a key role in the mechanism of neuropathic pain. Inhibiting the expression and function of P2X4R can inhibit the central sensitization of neuropathic pain [18,19]. P2X7R activation leads to an increase in intracellular calcium ion concentration as well as the release of a large number of proinflammatory factors such as IL-1β and TNF-α [15,20,21].

A rat model of ION-CCI was used in the present study [22,23,24]. Primary cultured OECs were transplanted into the rats at the end of surgery. Behavioral tests, Western blotting, immunofluorescence assay, reverse transcription (RT)-quantitative polymerase chain reaction (qPCR), and other methods were used to evaluate the effects of postoperative treatment and changes in P2X7R level in the TG of the rats.

## 2. Materials and Methods

### 2.1. Ethics Statement

All the protocols for the animal experiment were approved by the Animal Care and Ethics Committee of the Medical School of Traditional Chinese Medicine University (Nanchang, China). All experimental procedures and protocols were approved by the Experimental Animal Ethics Committee of the Medical School of Nanchang University (Nanchang, China). All experimental procedures were in accordance with the National Institutes of Health Guidelines for the Care and Use of Laboratory Animals.

### 2.2. Experimental Animals

Thirty-six clean Sprague−Dawley (SD) rats of both sexes weighing 180–220 g and aged 7–8 weeks were provided by the Department of Laboratory Animal Science at the Jiangxi University of Traditional Chinese Medicine (Nanchang, China) for the study (license no. SCXK (Gan) 2018–0003). All animals lived in a constant environment of 24 ℃ before the experiment. Animals had sufficient food and water and were fed in a 12-h dark and 12-h light cycle environment.

### 2.3. Primary Culture of OECs

The olfactory bulbs of newborn SD rats were both removed under aseptic conditions and washed twice in precooled phosphate-buffered saline (PBS) (Boster, Wuhan, China) containing double antibody. Blood vessels and meningeal tissue on the olfactory bulb surface were carefully removed. The olfactory bulbs were cut to 1 mm^3^ size using ophthalmic scissors. The tissue samples were digested through mechanical blowing and with 0.25% trypsin (NCM Biotech, Suzhou, China) for 15 min. After the digestion was terminated, Dulbecco’s modified Eagle medium/Nutrient Mixture F-12 (DMEM/F12, Gibco, Beijing, China) containing 15% fetal bovine serum (FBS) (CellMax, Beijing, China) was added to the samples. The mixture was centrifuged and the cells were suspended in the medium. Subsequently, the cells were inoculated in a Petri dish and cultured in an incubator (5% CO_2_, 37 °C). Then, the cells were purified using the Nash differential adherent method. Suspended cells were transferred into a new culture dish 18 h after the first inoculation, and these cells were transferred into a new culture dish again at 18 h after the second inoculation [23,25]. The culture medium was subsequently changed every 1–2 days. Then, cell purity was determined after 10 days. 

### 2.4. Identification of OECs

Primary cultured OECs on Day 10 were seeded into 12-well plates for 12 h. After fixation with 4% paraformaldehyde at 37 °C for 20–30 min, the cells were washed with PBS and sealed with 5% bovine serum albumin (BSA, Boster) for 30 min. Then, the cells were incubated with rabbit anti-rat p75 antibody (1:1000, Boster) and mouse anti-rat glial fibrillary acidic protein (GFAP) antibody (1:1000, Boster) overnight in a refrigerator at 4 °C [10,26]. On the second day, the primary antibody was removed and the cells were washed, three times, with PBS buffer. Then, the cells were incubated in goat anti-rabbit CY-3 antibody (1:200, Boster) and goat anti-mouse fluorescein isothiocyanate (FITC) antibody (1:200, Boster) in a dark environment at room temperature for 1 h. Then, the cells were washed, 3 times, with PBS buffer. Then, the samples were sealed with anti-fluorescence quenching agent containing 4′,6-diamidino-2-phenylindole (DAPI) (Solarbio, Beijing, China). The glass slides were then prepared and observed under a laser confocal microscope (Olympus, Tokyo, Japan).

### 2.5. ION-CCI Rat Model

Two-month-old SD rats were anesthetized via intraperitoneal injection with 10% chloral hydrate (0.35 mL/100 g), after which samples of their skins were prepared in a sterile environment. The facial skin, muscles, and connective tissue of each rat were carefully separated, and the infraorbital nerve was exposed. Ligation was performed with 0–4 absorbable sutures at an interval of 1–2 mm [22,24,27,28]. The ligation degree was 1/4th the diameter of the nerve without affecting normal blood supply to the nerve. The muscle and skin were sutured layer by layer, and the rats were resuscitated at a suitable temperature. The 36 rats were randomly divided into three groups (*n* = 12 per group): sham, ION-CCI, and OEC groups. Rats in the sham group underwent only nerve exposure without ligation. After nerve ligation was performed, rats in the OEC group were injected with a suspension containing 1 × 10^5^ cells. The rats were raised in the same environment after they recovered from the anesthesia (Figure 1).

### 2.6. Animal Behavior Tests

A von Frey wire (BME-404, Tianjin, China) was used to measure the facial mechanical pain threshold of the rats. Tests were conducted one day before and on Days 1, 3, 7, 11, and 14 after the surgery. The rats were placed in a clear organic glass cage for 10–15 min, after which the von Frey wire was used to stimulate the tentacle pad on the operative side of the rats. Minimum force was used in the beginning of the test. Then, the bending force was gradually increased until the rats showed a significant positive reaction. The test interval was 5–10 s and the maximum test force was 10 g. The von Frey wire value obtained was considered to be the mechanical facial pain threshold when three out of five tests had positive reactions [27,29]. The data were collected and evaluated by an independent team.

### 2.7. Western Blotting

On the 14th day after operation, fresh TG tissue was collected and homogenized in precooled lysis buffer (Solarbio, Beijing, China) for 30 min. Then, the tissue homogenate was centrifuged (10,000 r/min, 15 min) and the supernatant was collected. The protein concentration was determined using a BCA assay kit, and then heated to 95 °C for 10 min. The protein was electrophoresed in 10% polyacrylamide gel (90 V 40 min and 120 V 50 min), and then transferred to PVDF membrane in transfer buffer (90 V 90 min). The PVDF membrane was put into 5% skimmed milk and sealed for 1 h. Rabbit anti-rat P2X7 antibody (1:8000, Alomone Labs, Jerusalem, Israel) and mouse β-actin antibody (1:2000, OriGene, Rockville, MD, USA) were added to the membrane, followed by incubation overnight in a refrigerator at 4 °C. After 12 h, the excess primary antibody was washed 3 times with TBST and the membrane was incubated with goat anti-mouse horseradish peroxidase (HRP) antibody (1:10,000, ZSGB Biotech, Beijing, China) and goat anti-rabbit HRP antibody (1:10,000, Boster, Wuhan, China). One and a half hours later, the PVDF membranes were washed with TBST, again for 3 times, and electrochemiluminescence (chemical hypersensitive luminescence) solution (NCM Biotech, Jiangsu, China) was added to the membrane for analysis. The photographs were imaged using a BIO-RAD ChemiDoc XRS+ (BIO-RAD, Hercules, CA, USA).

### 2.8. Double-Labeling Immunofluorescence Assay

Rats were anesthetized with 10% chloral hydrate normal saline solution (0.35 mL/100 g). On the 14th day after operation, normal saline and fresh 4% paraformaldehyde were perfused through the heart. Then, we removed TG, fixed it with 4% paraformaldehyde for 1 day, then transferred it to 20% sucrose solution for one day, and finally soaked for one day with 30% sucrose solution. Then, TG sections (10 µm) were prepared on a freezing microtome and placed on immunosorbent slides. The sections were washed, three times, with PBS buffer, sealed with 5% BSA at 37 °C for 30 min, and incubated overnight with rabbit anti-rat P2X7 antibody (1:8000, Alomone Labs, Jerusalem, Israel) and mouse anti-rat GFAP antibody (1:1000, Boster) in a wet box at 4 °C [28,30]. After 12 h, the slide was removed and washed, 3 times, with PBS buffer, then, the tissue sections were incubated in CY-3 goat anti-rabbit antibody (1:200, Boster, Wuhan, China) and FITC goat anti-mouse antibody (1:200, Boster) in a dark environment at room temperature for 1 h. Next, the sections were washed again, 3 times, with PBS buffer and sealed with anti-fluorescence quenching sealing tablets containing DAPI dye (Solarbio, Beijing, China). The photographs of the samples were taken with a laser confocal microscope (Olympus, Tokyo, Japan).

### 2.9. RT-qPCR

Rats were anesthetized with 10% chloral hydrate normal saline solution (0.35 mL/100 g). The TG on the operative side of the SD rats was removed and put into precooled PBS buffer. After homogenization, total RNA was obtained through TRIzol reagent (Trans, Beijing, China). The quantity and quality of total RNA were measured using a NanoDrop 2000 spectrophotometer (Thermo Fisher Scientific, MA, USA). The cDNA was obtained by using a reverse transcription kit (TIANGEN, Beijing, China). Real-time fluorescence qPCR was performed in a thermal cycler using a premixed SYBR dye kit (TIANGEN, Beijing, China). The temperature procedure was as follows: 95 °C for 15 min followed by 40 cycles of 95 °C for 10 s and 60 °C for 32 s. Specific primers for the fluorescence qPCR were purchased from Gene Company (P2X7R forward primer: 5′ CTGTGAAATCTTTGCCTGGTG 3′; P2X7R reverse primer: 5′TGTTTCTCGTAGTATAGTTGTGGC3′; β-actin forward primer: 5′ CCGGGACCTGACTGACTACCTCA3′; β-actin reverse primer: 5′ GGACTCGTCATACTCCTGCTTGCTG 3′ (General Biosystems, Anhui, China)). The expression level of the P2X7 gene in each group was determined by the 2^−^^ΔΔCT^ method. 

### 2.10. Statistical Analysis

Data are expressed as mean ± standard error of the mean (SEM). The data obtained were analyzed using the SPSS 17.0 software (IBM, Chicago, IL, USA). Differences in data were compared using one-way analysis of variance test. Statistical significance was considered at *p* < 0.05.

## 3. Results

### 3.1. Morphology and Purity of OECs in Culture

During the process of culture, OECs had a fusiform appearance with large and stereoscopic cell bodies. The cells grew rapidly in 15% FBS and DMEM/F12. The immunofluorescence assay showed that the OECs were double-labeled by GFAP and P75. The purity of the OECs reached more than 90% under different visual fields, indicating that the purity was at the level required for subsequent experiments (Figure 2).

### 3.2. Assessment of Mechanical Withdrawal Threshold (MWT)

The MWT data were collected one day before and 1, 3, 7, 11, and 14 days after the surgery. The decrease in MWT was not significant in the sham group. In contrast, the MWT in the ION-CCI and OEC groups decreased significantly after surgery; however, the threshold in the OEC group was significantly higher than that in the ION-CCI group from Day 7 (*p* < 0.05). Additionally, the threshold in the sham group was significantly higher than that in the ION-CCI group (*p* < 0.05). There was a slight difference in MWT between the sham and OEC groups (Figure 3).

### 3.3. P2X7R Protein Expression in the TG

After 14 days of normal feeding, the P2X7R protein expression in the TG changed significantly in each group. Image J was used to analyze the gray value of the WB results, and we compared the expression of P2X7R through β-actin. As compared with the sham (*p* < 0.05) and OEC (*p* < 0.05) groups, the ION-CCI group showed a significantly higher expression of P2X7R protein. There was no significant difference in P2X7R protein level between the OEC and sham groups (*p* > 0.05) (Figure 4).

### 3.4. Changes in P2X7R mRNA Expression

The mRNA level of P2X7R in the TG of rats in each group changed significantly 14 days after the operation. Specifically, the mRNA level of P2X7R in the ION-CCI group was significantly higher than that in the sham group. These results may have been caused by continuous secretion of proinflammatory factors and continuous increases in ATP levels around the cells after long-term nerve ligation. Furthermore, we found that the mRNA level of P2X7R was significantly lower in the OEC group than that in the ION-CCI group (*p* < 0.05). In contrast, there was no significant difference in the mRNA level of P2X7R between the OEC and sham groups (*p* > 0.05) (Figure 5).

### 3.5. Double-Labeling Immunofluorescence Assay of Frozen Sections

The P2X7R fluorescence intensity of each group was analyzed by Image J. P2X7R is mainly expressed by glial cells, especially astrocytes, in the nervous system. Therefore, P2X7R in the TG sections were co-labeled with GFAP for immunofluorescence detection [31,32]. P2X7R and GFAP were observed to be co-located in the TG sections of the rats. The fluorescence intensity of P2X7R in the ION-CCI group was significantly higher (*p* < 0.05) than that in the sham group. Results obtained on Day 14 showed that the fluorescence intensity of P2X7R in the OEC group was significantly lower than that in the ION-CCI group (*p* < 0.05). There was no significant difference in P2X7R expression between the sham and OEC groups (*p* > 0.05) (Figure 6).

## 4. Discussion

OECs are neuroglial cells with a strong regeneration function. They can secrete various nerve growth factors such as BDNF and GDNF, and their role in nerve repair has been widely recognized [10]. OECs have also been used by some doctors to treat spinal cord injury with good therapeutic outcomes [10,11,12,13]. Primary OECs were used in this study because primary culture cells are usually in a similar state as real in vivo cells and undergo a small degree of change. The primary culture had several types of cells; therefore, we performed P75 and GFAP double labeling on the OECs on Day 10 to verify cell purity [10,26]. We also developed a rat model of TN by inducing chronic nerve injury via ligation of the infraorbital nerve of SD rats. The ION-CCI rat model is an internationally recognized animal model of TN. Directional facial combing is a method to evaluate spontaneous neuropathic pain in rats. After the operation, the rats showed abnormal facial washing movements and sensory changes. This spontaneous facial pain in rats is similar to the symptoms in patients with TN [24,33,34,35]. We tested the MWT of the infraorbital nerve innervation region of the rats. We found that the mechanical threshold of the rats had decreased significantly in the ION-CCI group, indicating that the model had been successfully established [33,34,36].

OEC transplantation improved the MWT in rats with chronic sciatic nerve ligation injury [37]. Current clinical treatments for TN are not completely effective. This is due to the continuous use of drugs or the adverse effects that patients experience after surgery [6,38]. Consequently, there is an urgent need to find a better treatment for TN. In this study, we explored whether transplantation of OECs can improve the symptoms of TN.

Time is a critical factor in the development of pain. Cell transplantation has been found to have a good therapeutic effect on rats with neuropathic pain two weeks after the procedure [37]. In the present study, TG was removed from the operative side of each rat and subjected to Western blot, immunofluorescence, and RT-qPCR analyses. The results of the analysis of protein and nucleic acid levels showed that OEC transplantation resulted in a decrease in P2X7R expression in the TG. It is possible that the powerful regeneration ability of OECs repaired the disintegrated myelin tissue and improved the faulty conduction of neural electrical signals [10]. Thus, harmless stimuli could be transmitted through normal pathways without being recognized as pain signals. Another possible reason for the observed results is that OECs have an excellent anti-inflammatory effect. [39,40]. P2X7R is involved in the development of pain. In macrophages, when P2X7R is activated, pores in the cell membrane opened to release proinflammatory cytokines such as interleukin-1β to promote inflammatory responses [41]. Furthermore, P2X7R activated caspase–1 and caspase–3–mediated release of inflammatory cytokines through the NLR family pyrin domain-containing 3 signaling pathway [42,43]. Furthermore, the P2X7R–mediated nuclear factor-κB signal transducer and activator of the transcription 3 pathway has resulted in the alleviation of neuropathic pain symptoms in rats [44]. In the present study, the P2X7R expression in the experimental rats decreased after OEC transplantation. However, the mechanism underlying this decrease in P2X7R expression and the respective pathway involved are still unknown. Therefore, we hope to further explore these internal mechanisms in future studies. 

In the present study, we transplanted OECs to the ligation site of ION. Our results showed significant improvement in the behavior of the rats after OEC transplantation. The P2X7R expression in the OEC group was also lower than that in the ION–CCI group. We think that OEC transplantation may lead to the inhibition of P2X7R activation and reduce the pain of TN in rats. Therefore, it may be considered to be a new treatment option for TN. 

## Figures and Tables

**Figure 1 brainsci-12-00706-f001:**
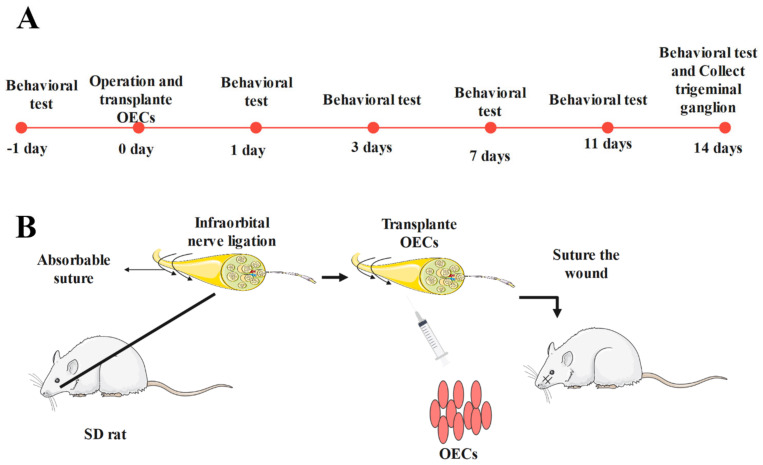
Experimental method: (**A**) Experimental timeline. In this part, we show the process of our animal experiment. Two to three days before the formal experiment, we conducted behavioral training on all animals to make them adapt to facial mechanical stimulation. Subsequently, behavioral tests were performed by an independent group on Days 1, 3, 7, 11, and 14 after operation; (**B**) ION–CCI operation in the rats and transplantation of OECs. For the operation, we anesthetized the rats and the infraorbital nerve was exposed carefully. After ligation, we transplanted OECs into the nerve surface and carefully sutured the muscle and skin to prevent the overflow of OECs.

**Figure 2 brainsci-12-00706-f002:**
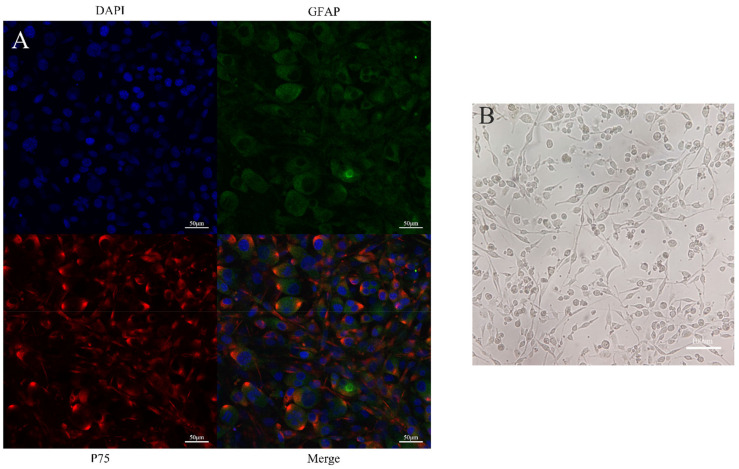
Primary identification of OECs: (**A**) The cells were identified via immunofluorescence analysis on Day 10 of the primary culture. The cell shape was fusiform, the cell body was round, and the two poles became filamentous. OECs are known to express both P75 and GFAP. The results of the immunofluorescence staining showed that most cells were stained red and green, which indicates a positive expression for P75 and GFAP. Number of positive cells: 93.37% ± 0.75. Scale bars, 50 μm; (**B**) OECs in growth state. The picture shows that the cell body of OEC is round, the poles are slender and spindle shaped, and the growth state is good. Scale bars, 100 μm. OECs, olfactory ensheathing cells; GFAP, mouse anti-rat glial fibrillary acidic protein.

**Figure 3 brainsci-12-00706-f003:**
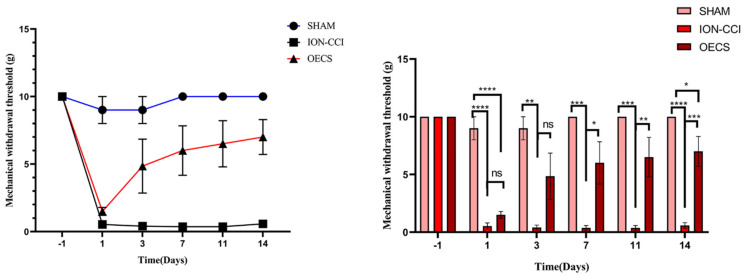
Mechanical withdrawal threshold (MWT). After the establishment of the chronic infraorbital nerve ligation injury model, the mechanical facial pain threshold in the sham group did not change significantly from the threshold value before the surgery was performed. Behavioral changes in the OEC group were noticeable from Day 3; but the changes were less noticeable in the ION-CCI group. The MWT was significantly lower in the ION-CCI group than the sham group (*p* < 0.05) on Day 14. However, the MWT was significantly higher in the OECs group than the ION-CCI group (*p* < 0.05) but still lower than the sham group (*p* < 0.05). The data were collected and evaluated by an independent team. Data are expressed as mean ± SEM (*n* = 4 per group, one-way analysis of variance followed by the least significant difference post hoc test). * Indicates *p* < 0.05; ** indicates *p* < 0.01; *** indicates *p* < 0.001; **** indicates *p* < 0.0001. ION-CCI, chronic constrictive injury of infraorbital nerve; MWT, mechanical withdrawal threshold; OEC, olfactory ensheathing cell; SEM, standard error of the mean.

**Figure 4 brainsci-12-00706-f004:**
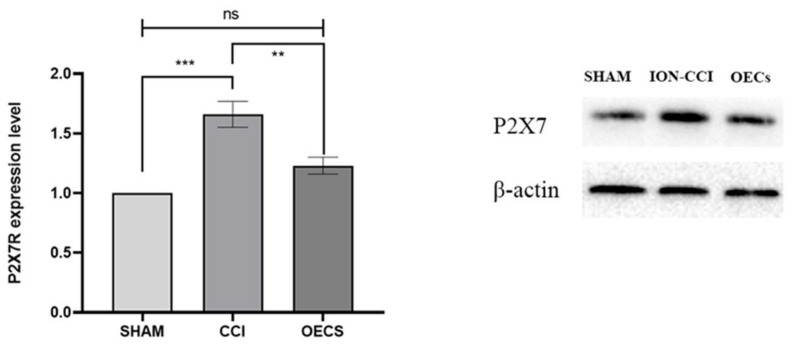
Changes in P2X7R protein level in TG. We used Image J to analyze the gray value of the WB results, and compared the expression of P2X7R through β-actin. The P2X7R expression was significantly higher in the ION-CCI group than in the sham group (*p* < 0.05). However, it was significantly lower in the OEC group than in the ION-CCI group (*p* < 0.05), and slightly higher in the OEC group than in the sham group (*p* > 0.05). Data are expressed as mean ± SEM (*n* = 4 per group, one-way analysis of variance followed by the least significant difference post hoc test). *** Indicates *p* < 0.001 when comparing the sham and ION-CCI groups, ** indicates *p* < 0.01 when comparing the ION-CCI and OEC groups, and Ns. means *p* > 0.05 when comparing the sham and OEC groups. The OEC group was comprised of rats with ION-CCI that were treated with OECs. ION-CCI, chronic constrictive injury of infraorbital nerve; OEC, olfactory ensheathing cell; SEM, standard error of the mean; TG, trigeminal ganglion.

**Figure 5 brainsci-12-00706-f005:**
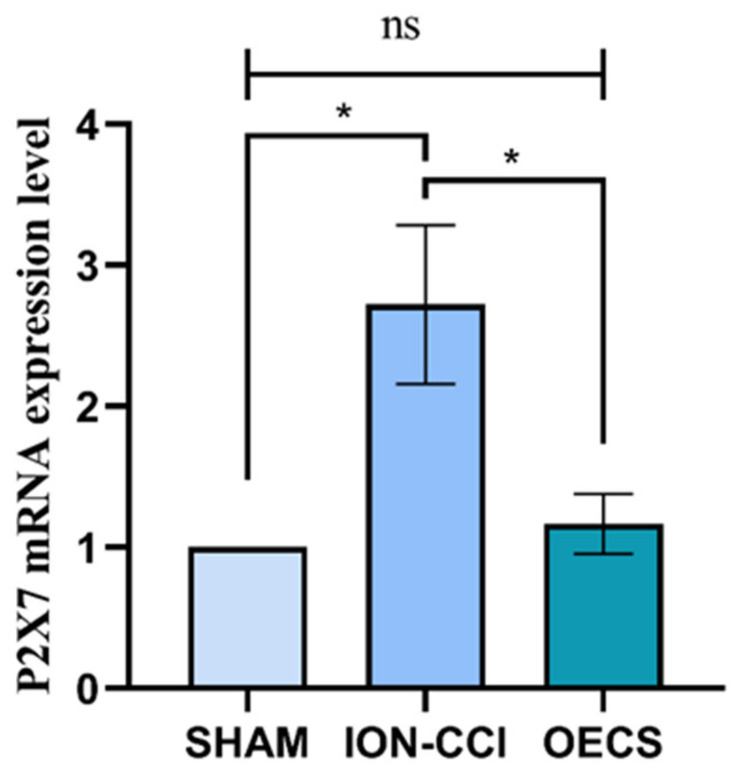
Changes of P2X7R mRNA level in TG. In this part, The expression level of P2X7R mRNA in each group was determined by the 2^−ΔΔCT^ method. The expression of P2X7R mRNA in the TG was significantly different among the groups. The mRNA level of P2X7R in the ION-CCI group was significantly higher than that in the sham (*p* < 0.05) and OEC (*p* < 0.05) groups. After OEC transplantation, the mRNA level of P2X7R in the TG of the rats decreased significantly and was similar to that in the sham group (*p* > 0.05). Data are expressed as mean ± SEM (*n* = 4 per group, one-way analysis of variance followed by the least significant difference post hoc test). * Indicates *p* < 0.05 when comparing the sham and ION-CCI groups and * indicates *p* < 0.05 when comparing the ION-CCI and OEC groups. Ns. indicates *p* > 0.05 when comparing the sham and OEC groups. The OEC group was comprised of rats with ION-CCI that were treated with OECs. ION-CCI, chronic constrictive injury of infraorbital nerve; OEC, olfactory. ensheathing cell; SEM, standard error of the mean; TG, trigeminal ganglion.

**Figure 6 brainsci-12-00706-f006:**
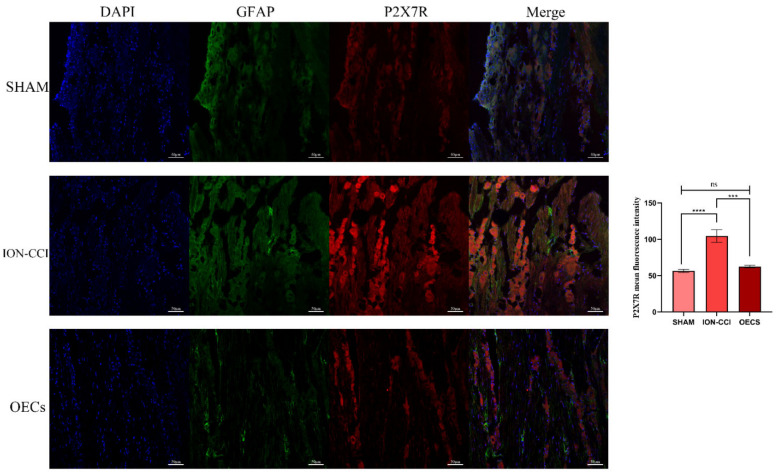
Double-labeling immunofluorescence assay. Frozen TG sections were labeled with P2X7R and GFAP and analyzed. The results showed that P2X7R and GFAP were co-expressed in the TG of the rats. The P2X7R fluorescence intensity in each group was quantitatively analyzed. We found that P2X7R (red) fluorescence intensity in the ION-CCI group was significantly higher than that in sham group (*p* < 0.05). Additionally, P2X7R fluorescence intensity in the OEC group was significantly lower than that in ION-CCI group (*p* < 0.05) and slightly higher than that in the sham group (*p* > 0.05). Scale bar, 50 μm. Data are expressed as mean ± SEM (*n* = 4 per group, one-way analysis of variance followed by the least significant difference post hoc test). **** Indicates *p* < 0.0001 when comparing the sham and ION-CCI groups, *** indicates *p* < 0.001 when comparing the ION-CCI and OEC groups, and Ns. indicates *p* > 0.05 when comparing the sham and OEC groups. The OEC group was comprised of rats with ION-CCI that were treated with OECs. GFAP, glial fibrillary acidic protein; ION-CCI, chronic constrictive injury of infraorbital nerve; OEC, olfactory ensheathing cell; SEM, standard error of the mean; TG, trigeminal ganglion.

## Data Availability

Datasets analyzed in the current study are available from the corresponding author on request.

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
