# Peer review of "Olfactory Ensheathing Cells Alleviate Facial Pain in Rats with Trigeminal Neuralgia by Inhibiting the Expression of P2X7 Receptor"

_brainsci, 2022, doi:10.3390/brainsci12060706_

Round 1

Reviewer 1 Report

In the aim to help finding a cure for trigeminal neuroplasia the authors claim to have establish a new model for this pathology by ligaturing the infraorbital nerve in rats referred to as ION CCI rat model. In this model the authors investigate both the validity of the test and the efficiency of recovery by following the mechanical withdrawal threshold and the expression levels of the P2PX7R protein using WB, qRT-PCR, and Immunofluorescence.

They also investigate the recovery potential of this pathology by injecting within the nerve constriction site purified primary OEC cells using similar methodological approaches.

Whereas the biological question asked is pertinent, the overall study is of poor quality, some strictly necessary controls are missing, the presented data are not sufficient to support the conclusions, and the study is poorly written to a point.

Important points

Description of the pathological model is not sufficiently presented. Histological sections showing the ligatured nerve and the subsequent damages on the myelinated fibers are necessary. Is not said whether other groups use similar techniques.

For the animal behavioral test, it is unclear whether such technique has been described by other groups. No reference is provided.

The authors make a clear correlation between expression of the protein/mRNA levels of the P2X7R protein and pain in the animal. Whereas such assumption might be correct it is not sufficient proof and further data using other markers should be presented. The authors do not present data for the P2XR4 protein supposedly another pain reporter for this pathology according to their introduction.

More importantly, the presented data show serious lack of methodology and some are not convincing at all. None of the strictly necessary internal controls are provided for the qPCR (page 5 and figure 4). The quantification methods are not described neither for the western blot (page and figure 3) nor for the immunofluorescence (page 5 and figure 5). The shown western blot is not convincing.

For the primary culture of OES the authors talk of a Nash differential adherent method without neither presenting it or giving a reference. The number of purified OECs cells should be counted, figure 1 "most cells were stained red or green".

The authors described a supposed recovery of the animal upon OECs injection. They present a behavioral test and not convincing diminution of expression of P2X7R data. The authors do not even look at P2X4R supposedly another pain reporter. These data are insufficient to support their conclusion.
The relevance of the model to the trigeminal neuralgia pathology is not discussed and the work of others in the field poorly described.

In conclusion this paper neither show that OEC cells alleviate facial pain in rats, neither that it is in a model mimicking trigeminal neuralgia, nor that OEC cells inhibit the expression of P2X7R.

Writing

The writing is overall to imprecise. Necessary information is in the discussion instead of the introduction. Important references are missing. Here are non-exclusive examples.

page 2

 "with good therapeutic outcomes." references?

Additionally, the threshold of the experimental group was significantly higher than that of the injured group (Nakai et al 2010). I dont understand this sentence nor what threshold the authors are referring to.

"A rat model of ION CCI was used in the present study."

There are no references. Is this the first time this model is presented? the model and its analyses are not discussed relative to the pathology nor relative to other models.

Page 9

"..with good therapeutic outcomes." references

"We found the MWT had decreased??? significantly indicating that the model had been successfully established". To me the MWT is increased in the model.

And others to long to list here

Author Response

Dear Reviewers:

Thank you for taking out of your busy schedule to review the manuscript. You have provided many valuable suggestions to our manuscript. Now we have carefully corrected and replied the manuscript for this revision. The revision instructions are as follows:

Point 1: Whereas the biological question asked is pertinent, the overall study is of poor quality, some strictly necessary controls are missing, the presented data are not sufficient to support the conclusions, and the study is poorly written to a point.

Response1: Thank you very much for your valuable comments. Indeed, in the original manuscript, we ignored many control details of the experiment, so we have corrected these errors in the revised manuscript. Here are some examples:

  1. page 2 “Experimental animals”

We supplemented the feeding conditions of experimental animals. For example, all animals lived in a constant environment of 24 ℃ before the experiment. Animals have sufficient food and water and were fed in a 12-hours dark and 12-hours light cycle environment.

  1. page 4 “Animal Behavior Tests”

This part of the data is collected by an independent team, so we believe that the experimental data is safe and reliable.

  1. page 4 “Western Blotting”

We provide the necessary data in the protein extraction process and the procedure of electrophoresis process. It is highlighted in the new manuscript.

  1. page 5 “Double-Labeling Immunofluorescence Assay”

We supplemented the key steps of immunofluorescence experiment. It is highlighted in the new manuscript.

  1. page 5 “RT-qPCR”

We provided the experimental program of RT qPCR and labeled it with highlight.

  1. page 2 paragraph 3

We have supplemented these parts that lack scientific citations. For example, ION-CCI rat model is a relatively mature animal model, which has been applied to the study of TN by many research groups [22,27]. The specific relationship between the model and TN will be explained later.

Reference 22: http://dx.doi.org/10.1097/j.pain.0000000000001521

Reference 27: 10.3389/fncel.2021.764141

Point 2: Description of the pathological model is not sufficiently presented. Histological sections showing the ligatured nerve and the subsequent damages on the myelinated fibers are necessary. Is not said whether other groups use similar techniques.

Response 2: Thank you for your suggestion. ION-CCI rat model is an internationally recognized animal model of TN, which can well simulate the spontaneous neuropathic pain symptoms of TN in rats. Directional facial combing is a method to evaluate spontaneous neuropathic pain in rats. After the operation, Rats showed abnormal facial washing movements and sensory changes. This spontaneous facial pain in rats is similar to the symptoms in patients with TN. We have improved the citation of ion-cci rat model.

According to the results of electron microscope, chronic ligation injury will affect myelinated fibers, and cell transplantation can alleviate it. However, as this part of the experimental results is more closely related to our follow-up research, I'm sorry that it can't be displayed in the manuscript for the time being. Here we provide some documents about ion-cci rat model for your reference:

Reference  43:10.1523/JNEUROSCI.14-05-02708.1994.

Reference  44:10.1016/s0304-3959(98)00039-6.

Reference  45:10.1124/jpet.103.050286.

Reference  46:10.3791/53167.

Point 3: For the animal behavioral test, it is unclear whether such technique has been described by other groups. No reference is provided.

Response 3: Following ION-CCI, rats also exhibit changes in response to tactile stimulation that are indicative of mechanical allodynia, and even very weak stimulus intensities provoke nocifensive behavior. Many research groups have adopted the von-frey method to evaluate the facial pain of rats. (reference 26,27,42)

Reference 26: http://dx.doi.org/doi:10.1016/j.neulet.2016.11.043

Reference 27: 10.3389/fncel.2021.764141

Reference 42: 10.3389/fncel.2021.672022

Point 4: The authors make a clear correlation between expression of the protein/mRNA levels of the P2X7R protein and pain in the animal. Whereas such assumption might be correct it is not sufficient proof and further data using other markers should be presented. The authors do not present data for the P2XR4 protein supposedly another pain reporter for this pathology according to their introduction.

    The authors do not even look at P2X4R supposedly another pain reporter. These data are insufficient to support their conclusion

Response 4: Your suggestion is very valuable. As we described in the manuscript, P2X4R has also been reported by many researchers to be involved in the pain process. This is our next research goal, and relevant experiments are ongoing.

Point 5: More importantly, the presented data show serious lack of methodology and some are not convincing at all. None of the strictly necessary internal controls are provided for the qPCR (page 5 and figure 4). The quantification methods are not described neither for the western blot (page and figure 3) nor for the immunofluorescence (page 5 and figure 5). The shown western blot is not convincing.

Response 5: Your suggestion has greatly improved the quality of the whole manuscript. According to your suggestion, we have supplemented the details of many experimental links, and highlighted them in the revised manuscript. For example, the specific control procedure of RT-qPCR should be as follows: 95 ° C for 15 minutes followed by 40 cycles of 95 ° C for 10 seconds and 60 ° C for 32 seconds. The specific experimental steps of Western bolt and immunofluorescence are also provided in the revised manuscript.

Point 6: For the primary culture of OES the authors talk of a Nash differential adherent method without neither presenting it or giving a reference. The number of purified OECs cells should be counted, figure 1 "most cells were stained red or green".

Response 6: Thank you for your comments. We have provided references for cell purification methods. (Reference 23,24) We counted the positive OECs in different visual fields, and calculated the percentage of positive cells by statistical method. (Number of positive cells: 93.37%±0.75.) At the same time, we provide bright field photos of cell culture. (Fig 2 B)

Point 7: The authors described a supposed recovery of the animal upon OECs injection. They present a behavioral test and not convincing diminution of expression of P2X7R data.

Response 7: Following ION-CCI, rats also exhibit changes in response to tactile stimulation that are indicative of mechanical allodynia, and even very weak stimulus intensities provoke nocifensive behavior. This means that the smaller the bending force that causes the facial stimulation response of rats, the more serious the symptoms of facial pain.

The purpose of our behavioral study is to prove the therapeutic effect of OECs transplantation. Behavioral data showed that the pain symptoms of rats after OECs transplantation were significantly reduced.

For the change of P2X7R expression, we mainly verified it by Western bolt, RT- qPCR and immunofluorescence.

Point 8: The relevance of the model to the trigeminal neuralgia pathology is not discussed and the work of others in the field poorly described.

Response 8: Thank you for your suggestion. We have described the ion-cci rat model in the revised manuscript and provided specific references to support it. ION-CCI rat model is an internationally recognized animal model of TN, which can well simulate the spontaneous neuropathic pain symptoms of TN in rats. Directional facial combing is a method to evaluate spontaneous neuropathic pain in rats. After the operation, Rats showed abnormal facial washing movements and sensory changes. This spontaneous facial pain in rats is similar to that in patients with TN. (Reference 43,44,45,46)

Reference 22: http://dx.doi.org/10.1097/j.pain.0000000000001521

Reference 27: 10.3389/fncel.2021.764141

Reference 42: 10.3389/fncel.2021.672022

Reference  43:10.1523/JNEUROSCI.14-05-02708.1994.

Reference  44:10.1016/s0304-3959(98)00039-6..

Reference  45:10.1124/jpet.103.050286.

Reference  46:10.3791/53167.

Point 9: Writing

Response 9: In this part, we added many references to the revised manuscript to support our view. At the same time, we also revised the content of our article. For example, we revised reference 14 and rewritten new ideas. Other modifications are also highlighted in the revised manuscript. Your comments have greatly improved the quality of our manuscripts.

To sum up, we have made necessary and detailed modifications to the article. Please do not hesitate to contact us if there are any question. Thanks again to you for your hard work! Best wishes to you!

Yours sincerely,

Reviewer 2 Report

Summary

The study presented by Lu et al describe the transplantation of olfactory ensheathing cells (OECs) into a rat model of trigeminal neuralgia and report a reduced expression of the purinreceptor P2X7 in trigeminal ganglia potentially leading to an increased mechanical pain threshold.

General Comments

  • Please use high resolution images

Major Comments

  • Please use high resolution images of the cells in figures 1 and 5. The provided images are very small and the specific stainings are not very well to see.
  • A brightfield image of the OECs would be nice to have to see the morphology of the cells. Did you also stain for other OEC marker proteins like Nestin or Vimentin? If so, please add. You state, that the cell population was 90% pure. Did you perform a FACS analysis? I would recommend to perform a FACS analysis using some of the OEC-specific marker proteins to really prove purity of the population.

Minor Comments

  • RNA and protein expression of P2X7R was analyzed in the isolated TG. How can a cross-contamination with other cell types be excluded? Did you use TG-specific marker genes/proteins for validation?
  • In figure 5 you show only colocalization of P2X7R with GFAP but no TG-specific marker was used. This would imply that after injury infiltrating glial/macrophage-like cell types upregulate P2X7R thereby inducing some kind of inflammatory conditions. What type of cell is expressing GFAP and do you have data on inflammatory processes in the tissue? If so, please add.

Author Response

Dear Reviewers:

Thank you for taking out of your busy schedule to review the manuscript. Now we have carefully corrected and replied the manuscript for this revision. The revision instructions are as follows:

Point 1: Please use high resolution images. Please use high resolution images of the cells in figures 1 and 5. The provided images are very small and the specific stainings are not very well to see.

Response 1: Thank you for your suggestion, we have replaced the pictures in the manuscript and provided larger and higher definition pictures. (Fig. 2; FIG. 6 for details)

Point 2: A brightfield image of the OECs would be nice to have to see the morphology of the cells. Did you also stain for other OEC marker proteins like Nestin or Vimentin?

Response 2: Thank you for your suggestion, we have provided bright field photos of OECs in their living conditions in the manuscript (Fig 2 B). OECs will change into two forms in cell culture. We mainly hope to obtain Schwann-cell-like OECs. According to the existing literature, this OECs mainly express GFAP and p75,at the same time most studies also use p75 as an excellent marker for OECs identification. (Reference 25,31 for details)

Reference 25: doi:10.1016/j.expneurol.2010.08.020

Reference 31: 10.1002/glia.23282

Point 3: You state, that the cell population was 90% pure. Did you perform a FACS analysis? I would recommend to perform a FACS analysis using some of the OEC-specific marker proteins to really prove purity of the population.

Response 3: Thank you for your suggestion, we have used immunofluorescence double labeling method to determine the purity of OECs (P75 and GFAP are specific markers of the cell types we need.). We took fluorescent films under different visual fields through cell climbing films, counted the positive cells under different visual fields, and made statistical analysis. It was found that the number of positive cells of OECs under different visual fields was greater than 90% (93.37%±0.75). And I'm sorry we didn't use FACS analysis.

Reference 25: doi:10.1016/j.expneurol.2010.08.020

Reference 31: 10.1002/glia.23282

Point 4: RNA and protein expression of P2X7R was analyzed in the isolated TG. How can a cross-contamination with other cell types be excluded?

Response 4: Here, we used mouse anti rat GFAP monoclonal antibody to ensure the specificity of the experimental results, and went through sufficient cleaning after each antibody incubation stage. Although we avoided other cell staining as much as possible, some nonspecific staining is still inevitable.

Point 5: Did you use TG-specific marker genes/proteins for validation?

Response 5: As for the staining of TG specific markers, we extracted the TG of rat skull base when extracting animal tissue materials, and there is no problem of wrong tissue source. At the same time, other literatures also use immunofluorescence labeling methods similar to ours (Reference 42 for details). Your question is a valuable opinion. In our subsequent experimental research, we will carry out specific staining for TG to eliminate some errors.

Reference 42: 10.3389/fncel.2021.672022

Point 6: In figure 5 you show only colocalization of P2X7R with GFAP but no TG-specific marker was used. This would imply that after injury infiltrating glial/macrophage-like cell types upregulate P2X7R thereby inducing some kind of inflammatory conditions. What type of cell is expressing GFAP and do you have data on inflammatory processes in the tissue?

Response 6:

Thank you very much for your valuable comments. We have modified the picture sequence of the manuscript, and Fig5 has been changed to Fig6. In TG, GFAP was mainly expressed by glial cells.

For your question about inflammation, we think this is a very attractive point. After we found that the expression of P2X7R in TG decreased after OECs transplantation, we carried out the study of related inflammatory factors. At present, the research is in progress. I'm very sorry that we can't provide relevant experimental data to you. We will show it in the next research report.

To sum up, we have made necessary and detailed modifications to the article. Please do not hesitate to contact us if there are any question. Thanks again to you for your hard work! Best wishes to you!

Yours sincerely,

Reviewer 3 Report

The authors used a rat ION-CCI of the infraorbital nerves as model for trigeminal neuralgia. In one group primary cultured OECs were transplanted into the rats at the end of surgery. Behavioral tests, western blotting, immunofluorescence assay, reverse transcription (RT)–quantitative polymerase chain reaction (qPCR), were used to evaluate the effects of transplantation of OECs in the ION-CCI model rats and changes in P2X7R level in the TG of the rats.  The authors found that the mechanical threshold was increased in the ION-CCI animals that had transplanted cells compared to the ION-CCI animals and this threshold was similar to the sham-treated control at 14 after treatment. They also demonstrated an increased expression of the P2X7R trigeminal ganglion.

A few comments are listed below:

There are a number of places where citations are missing

Citations are need for the latter part of Paragraph 1 page 2. 

More references are needed in the discussions.

Figure 1 From the figure shown it does not appear that 90% of the cells express both P75 and GFAP. 

Why does figure 2 come before figure 1?

Figure 2C.  Were there any statistical differences in the time points apart from day 14?  If yes that should be noted on the graphs

Figure 5.  Better representative images are needed. Need higher quality images as it is not clear whether GFAP and P2X7R are co-localized in the TG. There appears to be GFAP staining in the cell bodies of the neurons where it should not be. Is P2X7R located on the neurons or glial cells or both? Was there an increase in co-localization of P2X7R and GFAP in the ION-CCI TGs

The authors made the following statement; “however, OECs can alleviate local inflammation by secreting anti-inflammatory factors” – What anti-inflammatory factors are released and where is your citation for this statement.

The authors could have and maybe should characterize the OECs further to determine what is secreted from them.  Also, are the transplanted OECs still alive at 14 days? Where are they located? Is there evidence that these OECs cells are ensheathing the ligated nerve?  Maybe some histology on the ligation site would shed more light on these questions. 

Author Response

Dear Reviewers:

Thank you for taking out of your busy schedule to review the manuscript. Now we have carefully corrected and replied the manuscript for this revision. The revision instructions are as follows:

Point 1:

  1. There are a number of places where citations are missing

2.Citations are need for the latter part of Paragraph 1 page 2.

3.More references are needed in the discussions.

Response 1: Thank you very much for your valuable comments. We have supplemented and replaced many more convincing references in the manuscript. (Reference 28,29,30 in Paragraph 1 page 2) We added nearly 20 more convincing references.   

Point 2: Figure 1 From the figure shown it does not appear that 90% of the cells express both P75 and GFAP.

Response 2: Thank you for your valuable comments. We have modified the picture order of the manuscript. At present, the picture of cell identification is in Fig. 2. As for the number of positive cells, we think it may be due to the low resolution and unclear image. Here, we changed the picture with higher resolution, and we provided the statistical results of the number of OECs in different visual fields. (Number of positive cells: 93.37%±0.75)

Point 3: Why does figure 2 come before figure 1?

Response 3: Thank you for your advice. We have modified the picture order in the manuscript. At present, Fig1 is about the experimental process, and Fig2 is about the data of cell identification.

Point 4: Were there any statistical differences in the time points apart from day 14?  If yes that should be noted on the graphs

Response 4: The data on behavior has been shown in Fig. 3 of the new manuscript. As you said, at other time points, the facial behavioral data of rats in each group were statistically significant. The relevant statistical differences have been shown in the statistical chart.

Point 5: Better representative images are needed. Need higher quality images as it is not clear whether GFAP and P2X7R are co-localized in the TG. There appears to be GFAP staining in the cell bodies of the neurons where it should not be. Is P2X7R located on the neurons or glial cells or both? Was there an increase in co-localization of P2X7R and GFAP in the ION-CCI TGs.

Response 5: According to your suggestion, we have changed the pictures in the original manuscript and provided clearer pictures at the same time.

     Here, we used mouse anti rat GFAP monoclonal antibody to ensure the specificity of the experimental results, and went through sufficient cleaning after each antibody incubation stage. Although we avoided other cell staining as much as possible, some nonspecific staining is still inevitable.

According to the current literature data, P2X7R is mainly coexpressed with GFAP. P2X7R expression in glial cells is recognized. We provided two references, and their P2X7R staining results in TG and other peripheral ganglia were similar to ours. (Reference 41, 42)

The questions you raised are worth discussing in the follow-up experiments. In the future, we will further study the content of relevant factors.

Reference 41: 10.3389/fpsyt.2019.00770

Reference 42: 10.3389/fncel.2021.672022

Point 6: The authors made the following statement; “however, OECs can alleviate local inflammation by secreting anti-inflammatory factors” – What anti-inflammatory factors are released and where is your citation for this statement.

Response 6: Thank you very much for helping us point out the mistakes. We have revised this sentence, marked it with highlighted symbols in the original text, and provided convincing references. At present, articles have pointed out that OECs transplantation can reduce the inflammatory response, but there is no clear case of what causes it. (Reference 35, 36) We are studying the related factors.

Reference 35: 10.3389/fncel.2019.00341

Reference 36: 10.1007/s12035-016-9709-5

Point 7: The authors could have and maybe should characterize the OECs further to determine what is secreted from them.  Also, are the transplanted OECs still alive at 14 days? Where are they located? Is there evidence that these OECs cells are ensheathing the ligated nerve?  Maybe some histology on the ligation site would shed more light on these questions.

Response 7: Thank you very much for your advice. As you said, whether the cells survive after transplantation is very important. Our study found that cells can survive on the graft surface after transplantation. At the same time, histological and electron microscopic results showed that CCI could denature myelinated fibers.

    For the active substances you mentioned, another team of our team is studying the effect of OECs exosomes. The specific substances of OECs that down regulate the expression of P2X7R still need to be further studied.

For the above reasons, we intend to publish these data as our new research results, so we have not shown it in this manuscript for the time being.

To sum up, we have made necessary and detailed modifications to the article. Please do not hesitate to contact us if there are any question. Thanks again to you for your hard work! Best wishes to you!

Yours sincerely,